# Red blood cell-derived semaphorin 7A promotes thrombo-inflammation in myocardial ischemia-reperfusion injury through platelet GPIb

David Köhler[1], Tiago Granja[1], Julia Volz[2], Michael Koeppen[1], Harald F. Langer[3], Georg Hansmann[4], Ekaterina Legchenko [4], Tobias Geisler[5], Tamam Bakchoul [6], Claudia Eggstein[1], Helene A. Häberle[1], Bernhard Nieswandt[2] & Peter Rosenberger[1]✉

Myocardial ischemia is one of the leading health problems worldwide. Therapy consists of the restitution of coronary perfusion which is followed by myocardial inflammation. Platelet–neutrophil interaction is a crucial process during inflammation, yet its consequences are not fully understood. Here, we show that platelet–neutrophil complexes (PNCs) are increased in patients with acute myocardial infarction and that this is associated with increased levels of neuronal guidance protein semaphorin 7A (SEMA7A). To investigate this further, we injected WT animals with Sema7a and found increased infarct size with increased numbers of PNCs. Experiments in genetically modified animals identify Sema7a on red blood cells to be crucial for this condition. Further studies revealed that Sema7a interacts with the platelet receptor glycoprotein Ib (GPIb). Treatment with anti-Sema7a antibody protected from myocardial tissue injury. In summary, we show that Sema7a binds to platelet GPIb and enhances platelet thrombo-inflammatory activity, aggravating post-ischemic myocardial tissue injury.

[1] Department of Anesthesiology and Intensive Care Medicine, University Hospital, Tübingen, Germany. [2] Institute of Experimental Biomedicine and Rudolf Virchow Center, Würzburg, Germany. [3] Department of Cardiology, University Hospital Lübeck, Lübeck, Germany. [4] Department of Pediatric Cardiology, Hannover Medical School,  Lübeck, Germany. [5] Department of Cardiology, University Hospital, Tübingen, Germany. [6] Center for Clinical Transfusion Medicine, University Hospital of Tübingen, Tübingen, Germany. ✉email: peter.rosenberger@medizin.uni-tuebingen.de

Myocardial infarction (MI) remains one of the leading health disorders worldwide. In the treatment of MI, early reperfusion of the myocardium is the most effective therapy to improve clinical outcome[1]. However, reperfusion of the previously ischemic myocardium may also induce injury to the tissue[2]. This phenomenon, termed reperfusion injury (RI), reduces the beneficial effects of early reperfusion and is characterized by the infiltration of immune cells, mainly neutrophils, into previously ischemic areas, where they contribute to injury through tissue inflammation[3–6]. Platelets are increasingly recognized as central orchestrators of inflammatory processes, mainly by enhancing immune cell recruitment and modulation of endothelial barrier function, a phenomenon referred to as thrombo-inflammation[7]. In the course of myocardial IR (ischemia reperfusion), the formation of platelet–neutrophil complexes (PNCs) aggravates inflammatory tissue injury and is thus a marker for tissue inflammation[8,9]. Given the clinical significance of a dysfunctional myocardium through reperfusion injury, a better understanding of this process and how it is guided is needed.

The platelet-specific membrane receptor complex glycoprotein Ib-IX-V (GPIb-IX-V), with its ligand-binding subunit GPIb, mediates initial platelet adhesion under conditions of high shear by binding to von Willebrand factor (vWF) immobilized at sites of endothelial damage or activation[10]. In addition, GPIb binds a number of other ligands, including P-selectin[11], Mac-1[12], and coagulation factors XI and XII[13,14], and mediates thrombo-inflammatory processes in different organs and disease settings[15], but the exact mechanisms are not fully understood. The infiltration of neutrophils into inflammatory tissue sites is influenced by neuronal guidance proteins (NGPs), a class of guidance cues that was originally identified in the developing human nervous system[16]. Among them is the family of semaphorins including semaphorin 7A (SEMA7A for human, Sema7a for murine protein), which plays key roles in the regulation of the immune response[17]. Sema7a was initially described in the context of axonal growth as a messenger protein involved in the guidance of synapse formation for the neuronal circuitry[18]. Subsequent work has shown that it enhances autoimmune encephalitis through T-cell-dependent cytokine production and that it can also increase the infiltration of neutrophils into sites of tissue hypoxia[19,20]. A role for Sema7a in the cardiovascular system has been described recently in atherogenesis[21]. Hu et al. showed that disturbed flow resulted in an induction of Sema7a on vascular endothelium and that this results in an increased expression of leukocyte adhesion molecules on the endothelial surface. Sema7a is also expressed in platelets, but its role in platelet function is unknown[22]. Given the expression pattern of Sema7a within the cardiovascular system, we hypothesized it might have a role in myocardial IR injury.

We report here that soluble SEMA7A is elevated in plasma of patients with acute MI, and that Semaphorin 7A holds significant impact on the extent of MIRI. We show that Sema7a promotes myocardial thrombo-inflammation and tissue damage by reinforcing platelet thrombotic activity and PNC formation through a platelet GPIb-dependent mechanism. Conversely, inhibition of Sema7a results in reduced myocardial IR injury and can be pursued as future strategy to reduce post-ischemic tissue damage.

## Results

**Patients with MIRI show increased PNCs and plasma SEMA7A.** The number of PNCs increases in myocardial tissue during MIRI[9]. The detrimental effects of PNCs have been demonstrated in other organs, such as the lung, where they increase inflammatory tissue damage, resulting in impaired organ function[8]. In an attempt to better understand the interaction of platelets and neutrophils and the formation of PNCs during inflammatory myocardial injury, we obtained blood samples from patients with active myocardial ischemia and tested them for the presence of PNCs by FACS analysis. We compared these samples to patients undergoing cardiac surgical operations, who are also exposed to reperfusion after extracorporal circulation (HLM). These patients did not show signs of active ischemia. In addition, we obtained blood from off-pump cardiac surgical operations without reperfusion injury as well as healthy controls. We found that the patients suffering from an acute MI showed significantly more CD42b-positive neutrophils (i.e., PNCs) compared to healthy controls and patients undergoing cardiac surgical procedures with or without extracorporal circulation. Notably, the platelets in the conjugates were fully activated, as shown by the marked integrin αIIbβ3 activation (PAC-1 binding) and CD62P exposure (Fig. 1a–d). Given that reperfusion of ischemic tissue triggers inflammation and our previous finding that SEMA7A holds significant pro-inflammatory capacity, we also measured soluble SEMA7A in the blood of these patients. Indeed, SEMA7A increased in the plasma of MI patients but not in any of the other tested patient groups (Fig. 1e, Supplementary Table 1). We also found that the SEMA7A expression on erythrocytes is not age dependent (Supplementary Fig. 1). Experiments in mice revealed that plasma Sema7a increased very rapidly during cardiac ischemia, with a significant elevation detected as early as after 1 min of reperfusion (Supplementary Fig. 2). SEMA7A was released from erythrocytes in response to shear stress or tissue hypoxia, both of which are present during myocardial ischemia reperfusion in the affected areas (Supplementary Fig. 3).

**Injection of Sema7a aggravates myocardial IR injury.** The profound increase in Sema7a in plasma of patients and mice with myocardial ischemia raised the possibility that it has a functional role in the progression of myocardial IR injury. To test this directly, we injected recombinant Sema7a (rmSema7a fusion protein, 1 μg/mouse prior to experiment) into WT animals and found that this resulted in markedly increased infarct size compared to appropriate IgG Fc control (rmIgG$_{2A}$ Fc)-injected animals (Fig. 2a, b). This finding correlated with increased troponin I, a marker of myocardial tissue damage. Histological sections revealed increased tissue injury in Sema7a-injected animals compared to Fc controls and a reduced number of PNCs in the tissue areas at risk (Fig. 2c, d). In cell culture experiments, we found no direct pro-apoptotic effect of human Sema7a (rhSEMA7A) on cardiomyocytes, as assessed by the activation of caspase 3 (Supplementary Fig. 4a, b). After observing more PNCs within the myocardial tissue, and knowing the detrimental effects these PNCs can have[8,9], we next used flow cytometry to test for the presence of PNCs in whole blood (gating strategy Supplementary Fig. 5a). We found markedly increased numbers of PNCs in the blood of mice injected with rmSema7a, and the conjugates (Fig. 2h) also displayed stronger signals for the platelet markers GPIb, CD62P, and activated integrin αIIbβ3 (JON/A) (Fig. 2e–g, Supplementary Fig. 7a, b). To test whether this would be reflected by PNCs infiltrating the area at risk (AAR), we extracted this area and determined the number of PNCs within the myocardial tissues for further analysis with flow cytometry (gating strategy Supplementary Fig. 5b). We found that more PNCs had extravasated into the myocardium in the AAR in the animals injected with rmSema7a and that the conjugates displayed significantly increased signals for GPIb, P-selectin, and activated integrin αIIbβ3 (JON/A) on neutrophils (Fig. 2i–l, Supplementary Fig. 7c, d). We also tested the influence of rmSema7a on the rolling of neutrophils and found that rmSema7a influenced the adhesion of neutrophils in vitro and in vivo

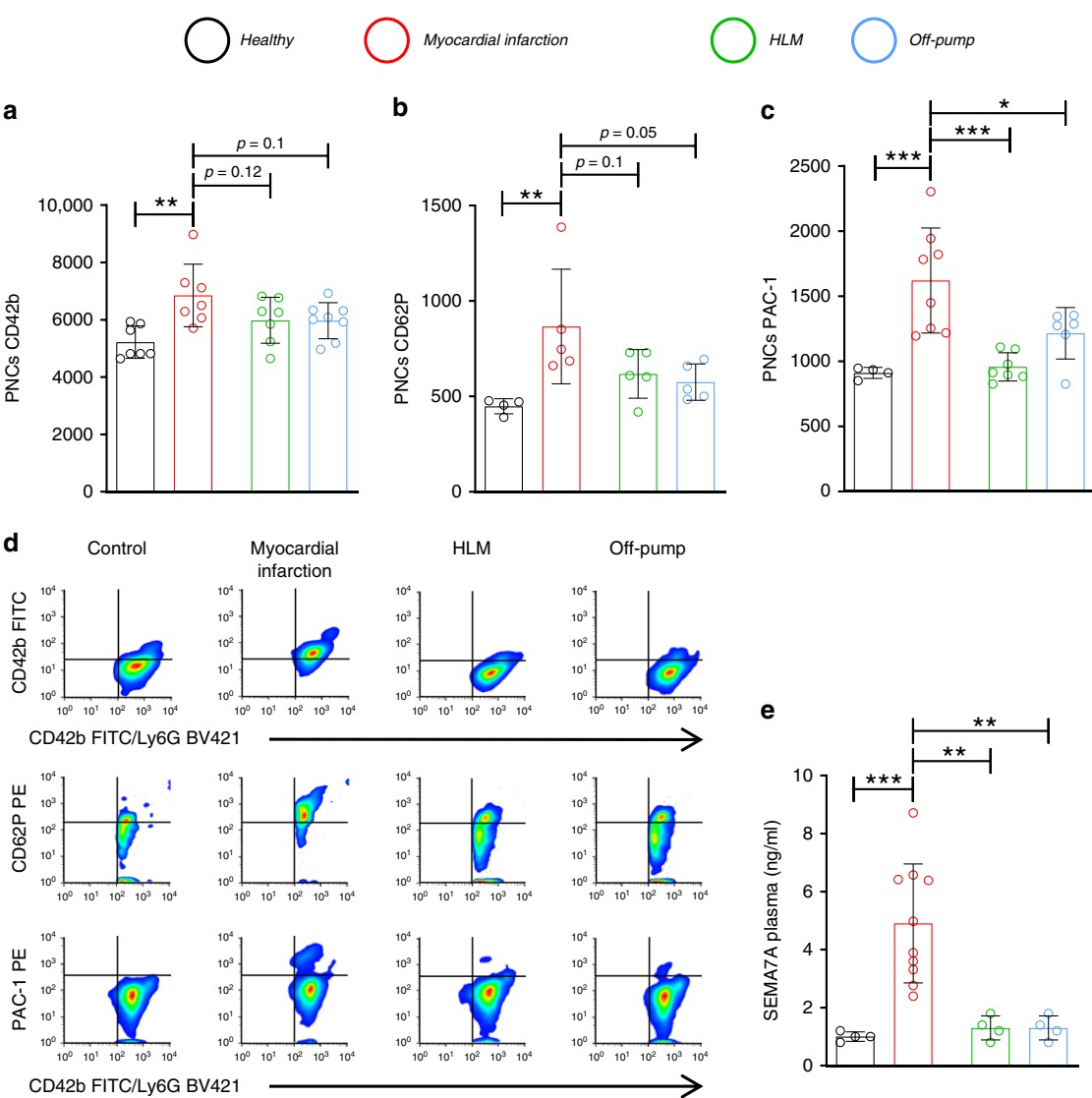

**Fig. 1 Patients with acute myocardial infarction show increased PNCs and plasma SEMA7A.** Samples were taken from patients with acute myocardial infarction, patients undergoing cardiac surgical operations with (HLM) or without cardiopulmonary bypass (Off-Pump) and healthy controls. **a** Flow-cytometric evaluation of PNCs' expression of GPIb (CD42b) in the presented patient groups (Healthy, MI, and HLM $n = 7$; Off-pump $n = 8$). **b** Flow-cytometric evaluation of PNCs for CD62P expression on neutrophils ($n = 4;5;5;5$). **c** PAC-1 binding to PNCs ($n = 4;8;7;6$). **d** Representative flow-cytometric histograms of groups measured and **e** Sema7a in the plasma of patients with acute MI ($n = 10$), patients undergoing cardiac surgical operations with (HLM) or without cardiopulmonary bypass (Off-Pump) and healthy controls (all $n = 4$/group). For comparison we performed one-way analyzes of variance followed by Dunnett's tests to group myocardial infarction. Data are mean ± SD; *$p < 0.05$, **$p < 0.01$, and ***$p < 0.001$ as indicated.

(Supplementary Fig. 6 and Supplementary Video 1). This is consistent with our previous findings that Sema7a expressed on the surface of endothelial cells increases the transmigration of neutrophils into inflamed tissues[19].

**Markedly reduced myocardial IR injury in Sema7a$^{-/-}$ mice.** To further investigate the role of Sema7a in myocardial IR injury, we employed Sema7a$^{-/-}$ animals and their littermate controls. Since Sema7a is involved in fibrotic transformation of tissues[23], we first assessed cardiac function in untreated WT and Sema7a$^{-/-}$ animals by dynamic magnet resonance tomography. We did not detect differences in the anatomy or cardiac performance parameters of Sema7a$^{-/-}$ mice versus littermate controls (Fig. 3a, b, Supplementary Fig. 8). Next, we exposed the Sema7a$^{-/-}$ mice to MIR and found that they developed dramatically smaller infarcts and displayed decreased plasma troponin I compared to littermate controls (Fig. 3c, d). Histological sections revealed decreased

tissue injury in Sema7a$^{-/-}$ animals compared to littermate controls and a reduced number of PNCs in the tissue areas at risk (Fig. 3e, f). Flow-cytometric analysis revealed decreased numbers of PNCs (Fig. 3j) in the blood of Sema7a$^{-/-}$ animals compared to controls, and the conjugates displayed lower levels of platelet activation markers (surface P-Selectin and activated integrin αIIbβ3 (JON/A)) in the early phase after IR (Fig. 3g–i, Supplementary Fig. 9a, b). We again tested whether this would be reflected within the PNCs in the AAR. We extracted this area and counted the PNCs within the myocardial tissues. Using flow cytometry, we found reduced number of PNCs within the AAR and the conjugates displayed decreased signals for GPIb, activated integrin αIIbβ3 and P-Selectin on neutrophils in the AAR of Sema7a$^{-/-}$ animals (Fig. 3k–n, Supplementary Fig. 9c, d).

**Red blood cell-derived semaphorin 7a is central to MIRI.** Next, we sought to identify the cellular source of soluble Sema7a that

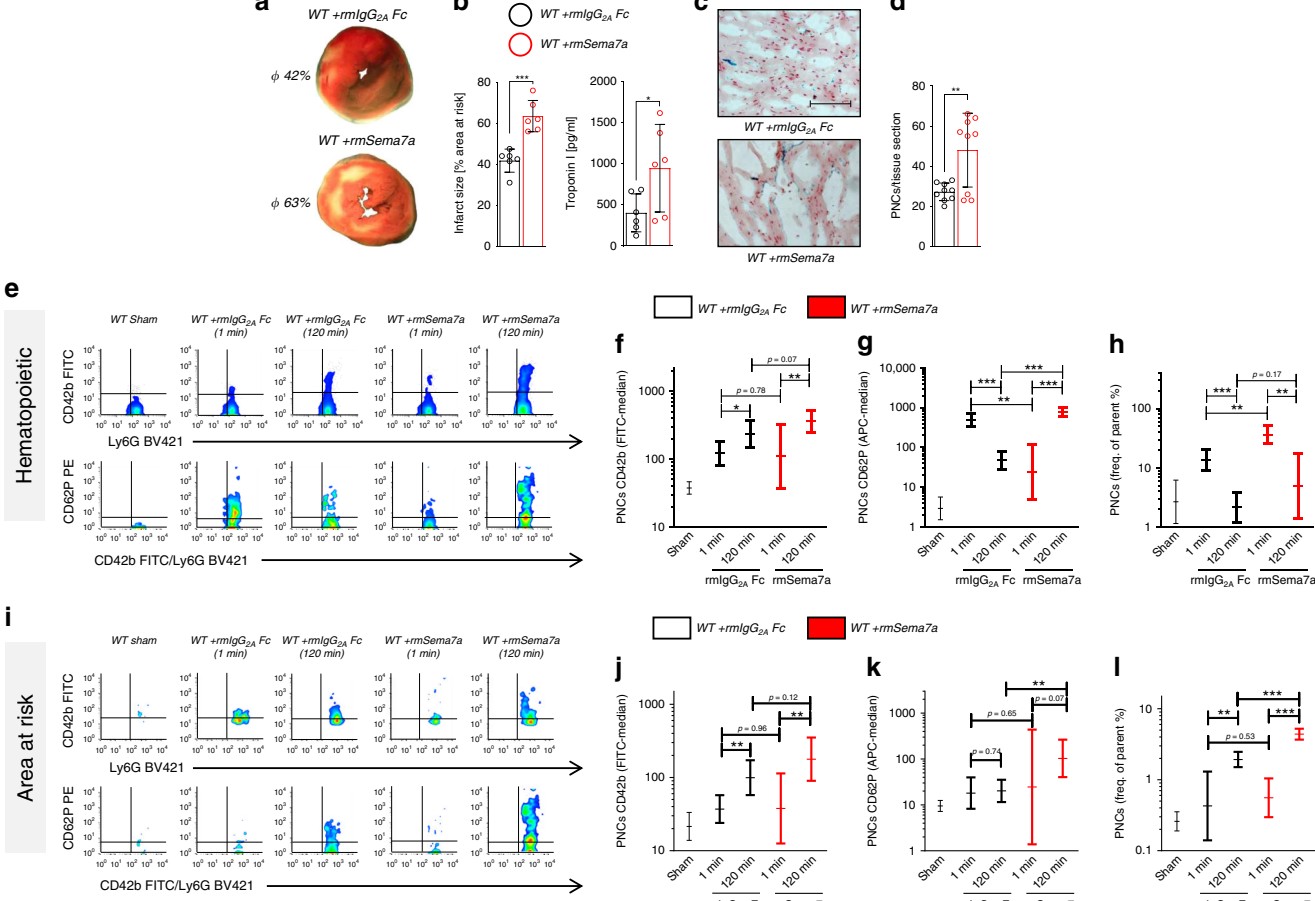

**Fig. 2 Injection of Sema7a results in increased MIRI and PNC formation.** Mice were injected with either recombinant semaphorin 7a (rmSema7a) or Fc control (rmIgG$_{2A}$ Fc) and then subjected to 1 h of ischemia followed by 2 h reperfusion. Samples were taken after 1 or 120 min reperfusion. **a** Representative TTC-stained slices of myocardial tissue showing infarcted area (blue/dark = retrograde Evans blue staining; red and white = AAR, white = infarcted tissue; 120 min) with **b** systematic evaluation of infarct sizes and corresponding troponin I plasma levels (120 min, $n = 6$/group). **c** Representative histological sections (scale bar 100 μm) of WT animals injected with either IgG Fc control or rmSema7a (120 min) and **d** number of PNCs counted in myocardial tissue sections of AAR in animals (120 min, $n = 9$/group). **e** Representative flow-cytometric plots of PNCs in the blood of sham animals, rmSema7a- or IgG Fc control–treated mice, showing GPIb (CD42b) and P-selectin (CD62P) after 1 and 120 min reperfusion. Systematic evaluation of flow-cytometric expression of mean fluorescence intensity (MFI) for **f** GPIb (CD42b, $n = 6;6;4;4;6$) and **g** P-selectin (CD62P, $n = 6;3;5;3;7$) expression. **h** Systematic evaluation of PNCs in % by flow cytometry in the blood of animals injected with rmSema7a or Fc control at 1 and 120 min ($n = 6;4;4;4;5$). **i** Representative flow-cytometric plots of PNCs in the AAR showing the presence of GPIb (CD42b) and P-selectin (CD62P) after 1 and 120 min reperfusion. Systematic evaluation of flow-cytometric expression of MFI for **j** GPIb (CD42b, $n = 5;5;5;4;6$) and **k** P-selectin (CD62P, $n = 6;4;5;3;5$) and **l** systematic evaluation of PNCs in % by flow cytometry in the AAR of animals injected with rmSema7a or Fc control at 1 and 120 min ($n = 5;4;5;5;5$). Comparisons in **b** and **d** were analyzed by unpaired two-tailed Student's $t$-tests (data are mean ± SD). For **f–h**, **j–l** we used log transformation of data to conform normality. For log-transformed data, unpaired two-tailed Student's $t$-tests were performed on the log values and results are displayed as geometric means and their 95% confidence intervals. *$p < 0.05$, **$p < 0.01$, and ***$p < 0.001$ as indicated.

mediated the observed pathogenic effect. Sema7a is expressed in several organs and tissues, with high abundance on red blood cells (RBCs) and only low expression within the myocardial tissue (Supplementary Fig. 10a–c). Therefore, we generated animals with genetic deletion of Sema7a in endothelial cells (Sema7a$^{loxP/loxP}$Tie2Cre+), cardiomyocytes (Sema7a$^{loxP/loxP}$Myh6Cre+), RBCs (Sema7a$^{loxP/loxP}$HbbCre+), and immune-competent cells (Sema7a$^{loxP/loxP}$LysMCre+) and subjected them to the MIRI model. In the Sema7a$^{loxP/loxP}$HbbCre+ animals, we found a significant reduction in infarct size, correlating well with reduced plasma TnI (Fig. 4a, b). Immunohistological analysis revealed a reduced number of PNCs within the myocardial tissue of these animals compared to WT control (Fig. 4c, d). Injection of rmSema7a showed that these results could be reversed in the Sema7a$^{loxP/loxP}$HbbCre+ animals although injection of Sema7a

resulted in higher troponin levels in these animals (Supplementary Fig. 11a–d). In the Sema7a$^{loxP/loxP}$Myh6Cre+ animals, we did not find a significant alteration of myocardial IR injury (Fig. 4e, f) and no changes compared to littermate controls when looking at the PNCs in the myocardial tissue (Fig. 4g, h). When exposing the Sema7a$^{loxP/loxP}$Tie2Cre+ animals to the same model, we found significantly smaller infarcts compared to controls (Fig. 4i, j). This protection was also reflected by a reduced number of PNCs, whereas the reduction in troponin I did not reach statistical significance (Fig. 4k, l). In contrast, infarct size was unaltered in Sema7a$^{loxP/loxP}$LysMCre+ mice compared to controls (Fig. 4m, n), as was troponin I level (Fig. 4o) and number of PNCs in the myocardial tissue (Fig. 4p). This approach showed that Sema7a expression on RBCs is critical to induce PNC formation and myocardial tissue injury.

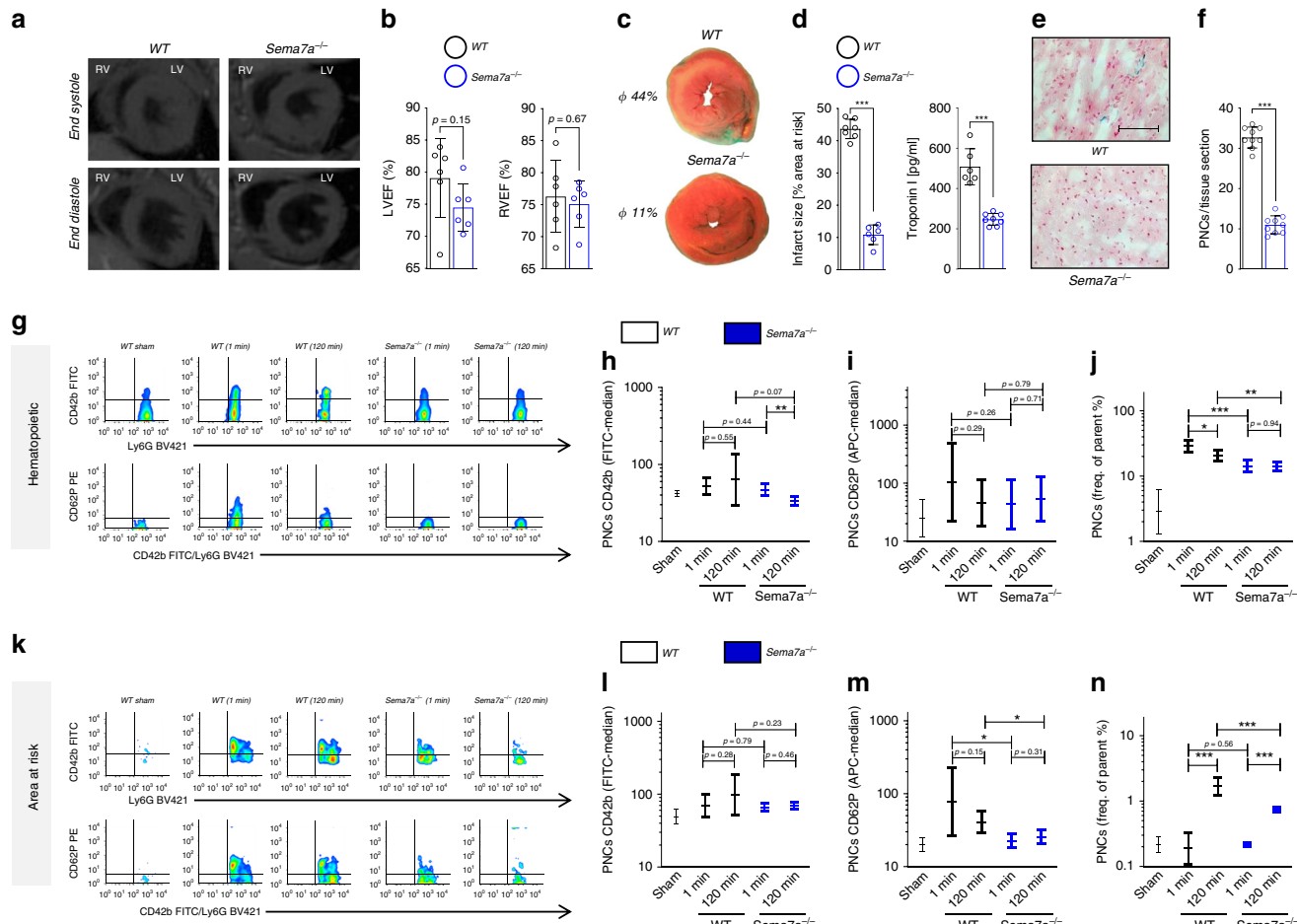

**Fig. 3 Sema7a$^{-/-}$ animals show reduced signs of MIRI and attenuated PNC formation. a** Sema7a$^{-/-}$ animals and littermate controls were evaluated by magnetic resonance tomography (MRT) to exclude anatomical and functional alterations in Sema7a$^{-/-}$ animals ($n = 6$/group). **b** Left and right ventricular ejection fraction (LVEF or RVEF in %) determined through MRT ($n = 6$/group). **c** Sema7a$^{-/-}$ mice and littermate controls were exposed to 60 min of ischemia and 120 min of reperfusion with samples taken after 1 or 120 min of reperfusion. Representative TTC-stained slices of myocardial tissue showing infarcted area (blue/dark = retrograde Evans blue staining; red and white = AAR, white = infarcted tissue, 120 min) with **d** systematic evaluation of infarct sizes ($n = 7;6$) and corresponding troponin I plasma levels ($n = 6;8$). **e** Representative histological sections (scale bar 100 μm) of Sema7a$^{-/-}$ and littermate control hearts (120 min). **f** Number of PNCs in myocardial tissue sections in AAR of Sema7a$^{-/-}$ mice and littermate controls (120 min, $n = 9$/group). **g** Representative flow-cytometric plots of PNCs in the blood of Sema7a$^{-/-}$ and littermate controls, showing GPIb (CD42b) and P-selectin (CD62P) after 1 and 120 min reperfusion. Mean fluorescence intensity (MFI) for **h** GPIb (CD42b, $n = 6;7;7;7;7$) and **i** P-selectin (CD62P, $n = 7$/group) in PNCs and **j** systematic evaluation of PNCs in % by flow cytometry in the blood of Sema7a$^{-/-}$ animals and littermate controls ($n = 7$/group). **k** Representative flow-cytometric plots of PNCs in the AAR and MFI for **l** GPIb (CD42b, $n = 7;7;7;7;6$) and **m** P-selectin (CD62P, $n = 7;6;7;7;7$) and **n** systematic evaluation of PNCs by flow cytometry in % in the AAR of WTs and Sema7a$^{-/-}$ animals at 1 min and 120 min ($n = 6;7;7;7;7$). Comparisons in **b**, **d**, **f** were analyzed by unpaired two-tailed Student's $t$-tests (data are mean ± SD). For **h–j**, **l–n** we used log transformation of data to conform normality. For log-transformed data, unpaired two-tailed Student's $t$-tests were performed on the log values and results are displayed as geometric means and their 95% confidence intervals. *$p < 0.05$, **$p < 0.01$, and ***$p < 0.001$ as indicated.

**Sema7a interacts with platelet glycoprotein Ib**. The above experiments have shown that Sema7a enhances platelet activation and PNC formation in the setting of myocardial IR. To test whether Sema7a directly acts on platelets, we first assessed its effects on platelet function in standard aggregometry. Unexpectedly, Sema7a did not induce any detectable platelet activation or aggregation at the concentration of 1 μg/ml. This was also confirmed by flow-cytometric analysis of platelet activation. Increasing concentrations of Sema7a had no effect on integrin αIIbβ3 activation (JON/A-PE) or P-selectin exposure under static conditions. In sharp contrast, a profound prothrombotic activity of Sema7a was observed when thrombus formation on collagen was assessed under flow using a whole-blood perfusion system. At a medium to high shear rate (1000 s$^{-1}$), reflecting arterial blood

flow, Sema7a markedly increased both surface area covered by platelets and thrombus volume (Fig. 5a, b), and the same effect was observed at a low shear rate (400 s$^{-1}$, Supplementary Fig. 12a, b). The flow-dependence of the prothrombotic Sema7a effect indicated a possible involvement of the vWF receptor, GPIb-IX, in this process, which is particularly important for thrombus formation under conditions of high shear[24]. To test this directly, we inhibited GPIb-IX by adding Fab fragments of the antibody p0p/B, which completely blocks the ligand-binding site on the GPIbα subunit of the receptor complex[15]. Under these conditions, the thrombus-promoting effect of Sema7a was lost completely at both low and high shear rates, demonstrating that it was GPIb-dependent (Fig. 5a, b). This result was further confirmed in mice in which the ectodomain of GPIbα was replaced

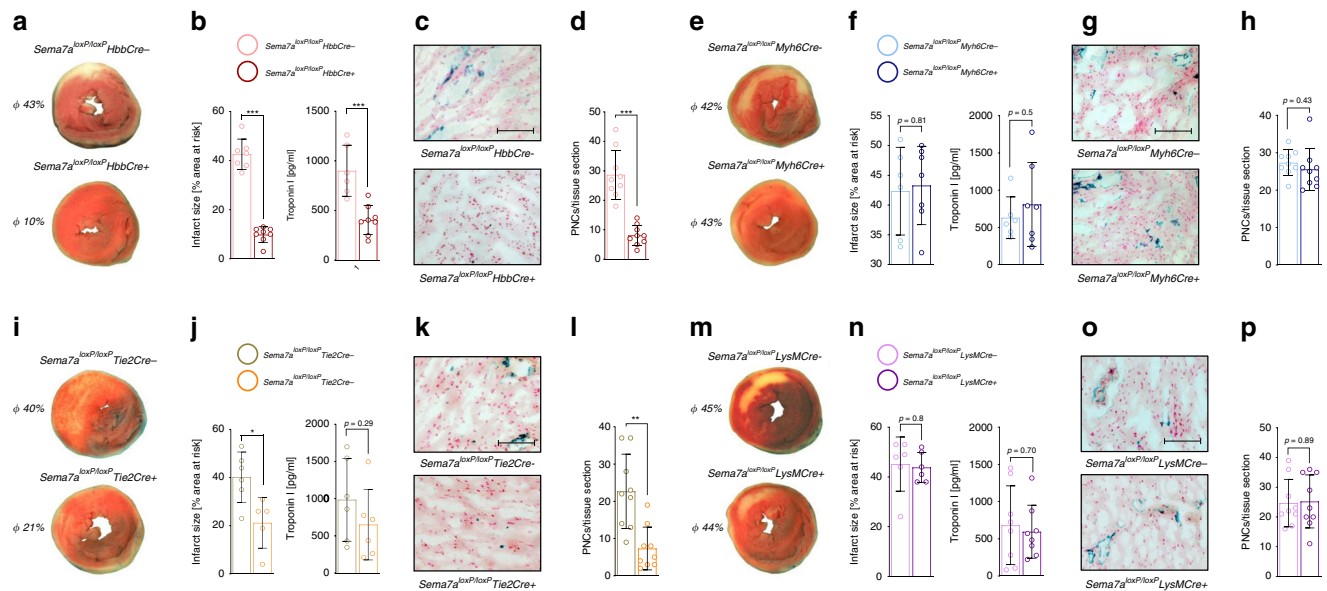

**Fig. 4 RBC-derived Sema7a drives PNC formation and aggravates MIRI.** *Sema7a^{loxP/loxP}HBBCre+*, *Sema7a^{loxP/loxP}Myh6Cre+*, *Sema7a^{loxP/loxP}Tie2Cre+*, and *Sema7a^{loxP/loxP}LysMCre+* animals or littermate controls were exposed to 60 min of ischemia and 120 min reperfusion. **a** Representative TTC-stained slices of myocardial tissue showing infarcted area (blue/dark = retrograde Evans blue staining; red and white = AAR, white = infarcted tissue) in *Sema7a^{loxP/loxP}HBBCre+* or littermate controls with **b** systematic evaluation of infarct sizes (*n* = 7;8) and correlating troponin I plasma levels (*n* = 6;8). **c** Representative histology sections (scale bar 100 µm) of *Sema7a^{loxP/loxP}HBBCre+* animals or littermate controls and **d** number of PNCs counted from myocardial AAR sections in *Sema7a^{loxP/loxP}HBBCre+* animals or littermate controls (*n* = 9/group). **e** Representative TTC-stained slices of myocardial tissue showing infarcted area in *Sema7a^{loxP/loxP}Myh6Cre+* or littermate controls with **f** systematic evaluation of infarct sizes and correlating troponin I plasma levels (*n* = 6;7). **g** Representative histological sections of *Sema7a^{loxP/loxP}Myh6Cre+* animals or littermate controls (scale bar 100 µm) and **h** number of PNCs counted in myocardial tissue sections of *Sema7a^{loxP/loxP}Myh6Cre+* animals or littermate controls (*n* = 9/group). **i** Representative TTC-stained heart slices showing myocardial infarcts in *Sema7a^{loxP/loxP}Tie2Cre+* or littermate controls with **j** systematic evaluation of infarct sizes (*n* = 6;5) and correlating troponin I plasma levels (*n* = 6;6). **k** Representative histology sections of *Sema7a^{lox/loxP}Tie2Cre+* animals or littermate controls (scale bar 100 µm) and **l** number of PNCs counted in myocardial tissue sections of *Sema7a^{loxP/loxP}Tie2Cre+* or littermate controls (*n* = 9/group). **m** Representative TTC-stained slices of myocardial tissue showing infarcted area in *Sema7a^{loxP/loxP}LysMCre+* or littermate controls with **n** systematic evaluation of infarct sizes (*n* = 6/group) and correlating troponin I plasma levels (*n* = 8/group). **o** Representative histology sections of *Sema7a^{loxP/loxP}LysMCre+* animals or littermate controls (scale bar 100 µm) and **p** number of PNCs counted from myocardial tissue sections of *Sema7a^{loxP/loxP}LysMCre+* or littermate controls (*n* = 9/group). All comparisons in this figure were analyzed by unpaired two-tailed Student's *t*-tests (data are mean ± SD). *$p < 0.05$, **$p < 0.01$ and ***$p < 0.001$ as indicated.

by the human IL-4 receptor (*GPIb-IL-4tg*) (Fig. 5c, d), where Sema7a did not induce an increase in surface coverage or thrombus volume in the whole-blood perfusion system.

To test whether the thrombo-inflammatory action of Sema7a myocardial IR injury also depends on platelet GPIb, we first subjected *GPIb-IL-4tg* animals to the MIRI model. Strikingly, these animals showed markedly reduced infarct size compared to WT controls, which was also reflected in the troponin I measurements. Furthermore, treatment of these mutant animals with Sema7a did not enhance MIRI, showing that this pathogenic activity of Sema7a was entirely GPIb dependent (Fig. 5e, f). When evaluating the tissue sections of the AAR of these animals, we found a low number of PNCs within myocardial tissue sections (Fig. 5g, h). To test a possible interaction of Sema7a with GPIb, we performed coimmunoprecipitation experiments. In a first step, we immunoprecipitated Sema7a from blood and myocardial tissue samples and blotted for the presence of GPIb. Indeed, GPIb coimmunoprecipitated with Sema7a in blood and especially the myocardial AAR in response to MIR. We then reversed this approach with immunoprecipitation of GPIb and blotting for Sema7a. Again we detected an interaction of Sema7a with GPIb in blood and AAR in response to MIR (Fig. 5i, j).

**Anti-Sema7a reduces PNC formation and MIRI.** To test whether inhibition of endogenous Sema7a affects MIRI, we injected a function-blocking anti-Sema7a antibody or IgG control (1 µg/

mouse) before the start of reperfusion. Indeed, anti-Sema7a treatment resulted in decreased infarct size and reduced troponin I compared to IgG control-injected animals (Fig. 6a, b). The histological sections of the AAR revealed decreased tissue injury in the anti-Sema7a-injected animals compared to IgG controls, with a reduced number of PNCs in the tissue areas at risk (Fig. 6c, d). Flow cytometry showed an increased number of CD42b-positive and P-selectin-positive neutrophils in the blood of the IgG-injected animals early on in the process after IR that was not observed in the anti-Sema7a-injected animals (Fig. 6e–g, Supplementary Fig. 13a, b). Further, we found a significantly reduced number of PNCs in the blood of these anti-Sema7a-injected animals (Fig. 6h). Analysis of the AAR revealed a marked increase in PNCs with fully activated platelets (evident by high signals for GPIb, activated integrin αIIbβ3 and P-selectin) at the later time point (120 min) in the IgG control-injected animals, which was virtually absent in anti-Sema7a-injected animals (Fig. 6i–k, Supplementary Fig. 13c, d), and a lower number of PNCs within the AAR (Fig. 6l). These data demonstrate that PNC formation has detrimental effects on myocardial tissue injury in experimental myocardial IR and can be effectively prevented by inhibition of endogenous Sema7a.

## Discussion
Myocardial ischemia followed by reperfusion remains one of the most significant health problems worldwide. Intervention to

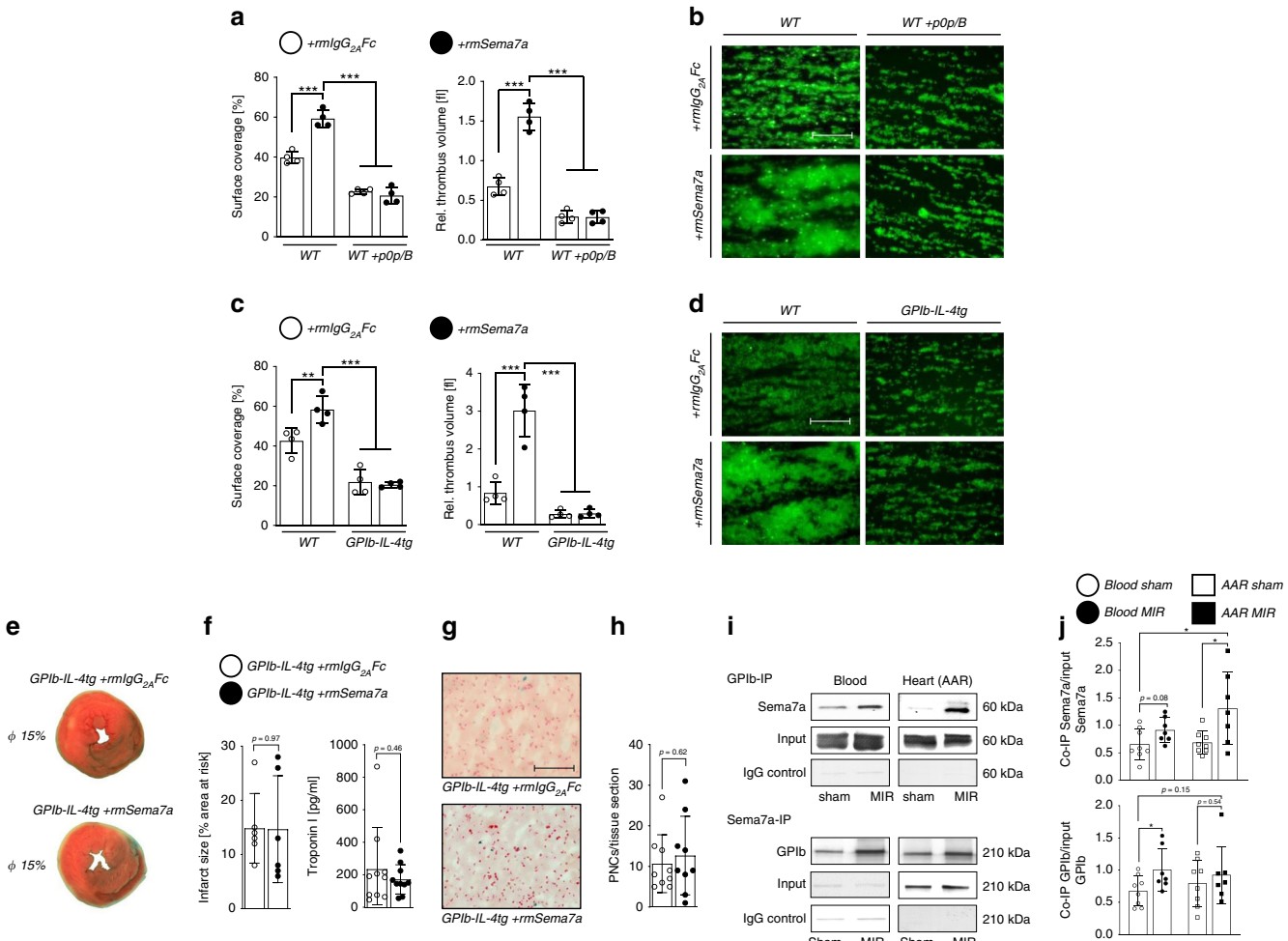

**Fig. 5 Sema7a exerts its function through platelet glycoprotein Ib (GPIb). a** rmSema7a markedly enhances adhesion and thrombus formation of WT platelets on collagen under flow at a shear rate of 1000 s$^{-1}$. Blockade of the ligand-binding site of platelet GPIb (p0p/B) abolishes this thrombus-promoting effect of rmSema7a ($n = 4$/group). **b** Representative IF images (scale bar 50 μm) of the experiment in **a**. **c** rmSema7a does not affect adhesion or thrombus formation of GPIb-IL-4-tg platelets on collagen under flow at a shear rate of 1000 s$^{-1}$ ($n = 4$/group). **d** Representative IF images (scale bar 50 μm) of the experiment in **c**. The representative fluorescence images as well as the mean surface coverage and relative thrombus volume are shown, as measured by integrated fluorescence intensity (IFI) per mm$^2$. **e** GP-Ib-IL-4-tg animals were exposed to 60 min of ischemia and 120 min of reperfusion. They were injected with rmSema7a or rmIgG$_{2A}$ Fc control right before ischemia. **f** Infarct sizes ($n = 6$/group) and correlating troponin I plasma levels were determined ($n = 10$/group). **g** Representative histology sections (scale bar 100 μm) and **h** number of PNCs counted from myocardial tissue sections of GP-Ib-IL-4-tg injected with either rmSema7a or rmIgG$_{2A}$ Fc (control, $n = 9$/group). **i** Co-immunoprecipitation analysis was performed between Sema7a and GPIb in blood and heart (AAR) tissue. Sema7a was affinity-precipitated using anti-GPIb antibody, and GPIb was affinity-precipitated using Sema7a antibody. As a negative control, rat or mouse IgG was used. Bound proteins were analyzed by immunoblotting. The same protein quantity was applied in all input loadings of blood or heart lysates. **j** Densitometric analysis was performed to quantify the immunoblots (data are mean ± SD; $n = 8;7;8;7$). In **a**, **c** we performed one-way analyzes of variance followed by Dunnett's tests to group WT +rmSema7a (data are mean ± SD). Comparisons in **f**, **h**, and **j** were analyzed by unpaired two-tailed Student's $t$-tests (data are mean ± SD). *$p < 0.05$, **$p < 0.01$, and ***$p < 0.001$ as indicated.

recanalize the occluded coronary artery is a crucial part of the initial therapy for this condition and significantly improves overall patient outcome[1]. Following occlusion, the subsequent reperfusion injury of the myocardium is the result of an inflammatory response that affects a large portion of patients with MI and can then result in severe myocardial dysfunction. Here, we show that the neuronal guidance protein semaphorin 7a released from the membrane of red blood cells is a mediator of inflammatory myocardial injury. We further provide evidence that Sema7a interacts with platelet GPIb and thereby promotes the formation of platelet–neutrophil complexes and their translocation into the affected myocardium, thereby increasing myocardial injury. To illustrate the role of Sema7a on platelets and the effect of Sema7a on MIRI we have provided a sketch in Fig. 7. Our results indicate that a strategy to interfere

with the Sema7a–GPIb interaction might result in reduced cardiac damage and improved myocardial outcome following MI and should therefore be pursued as a therapeutic option in the future.

GPIb-IX-V is a multifunctional and platelet-specific receptor complex with essential functions in the biogenesis, hemostatic functions and clearance of platelets. In addition, it is increasingly recognized that GPIbα is a key orchestrator of platelet–immune cell interactions and thrombo-inflammatory processes, such as experimentally induced IR injury in brain and liver in mice, which make the protein a promising therapeutic target. However, the mechanisms underlying these multiple functions of the GPIb-IX-V complex are largely elusive, mainly for two reasons. On the one hand, the intracellular signaling pathways and resultant cellular responses downstream of GPIbα have been difficult to study

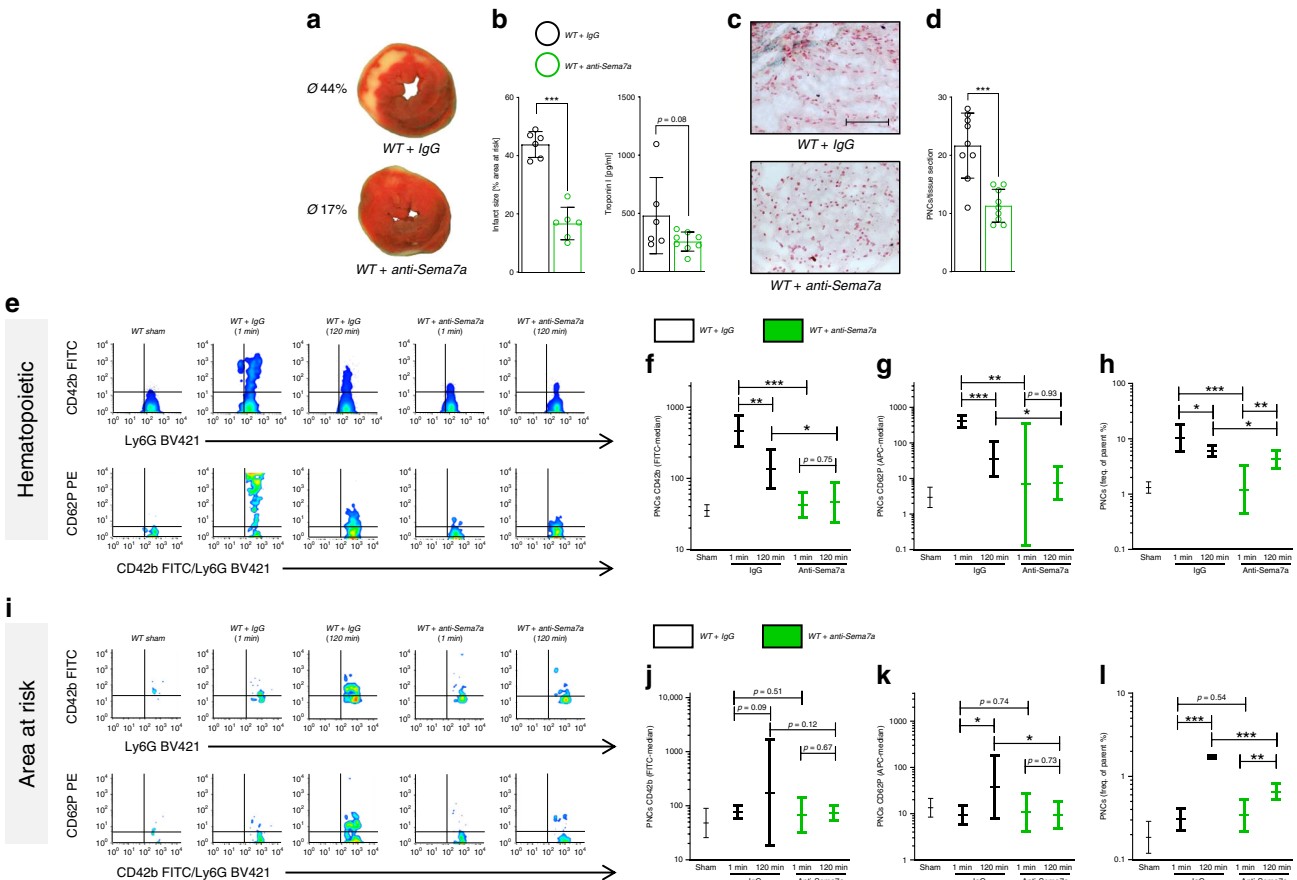

**Fig. 6 Anti-Sema7a reduces MIRI, platelet activation and PNC formation.** Animals were injected with either anti-semaphorin 7A antibody (anti-Sema7a) or IgG control 5 min before starting 120 min reperfusion after 60 min of ischemia, with samples taken after 1 or 120 min of reperfusion. **a** Representative TTC-stained heart slices of myocardial infarcts (blue/dark = retrograde Evans blue staining; red and white = AAR, white = infarcted tissue) with **b** systematic evaluation of infarct sizes ($n = 6$/group) and correlating troponin I plasma levels ($n = 6;8$). **c** Representative histology sections (scale bar 100 µm) of WT animals injected with either IgG control or anti-Sema7a and **d** number of PNCs counted from myocardial tissue sections ($n = 9$/group). **e** Representative flow cytometry plots of PNCs in the blood of Sham, anti-Sema7a-injected or IgG control-injected animals, expressing GPIb (CD42b) and P-selectin (CD62P). Systematic evaluation of flow-cytometric expression of mean fluorescence intensity (MFI) for **f** GPIb (CD42b, $n = 5;4;3;4;5$) and **g** P-selectin (CD62P, $n = 6;4;4;3;5$) and **h** systematic evaluation of PNCs in % by flow cytometry in the blood of animals injected with anti-Sema7a or IgG control at 1 and 120 min ($n = 3;4;4;3;4$). **i** Systematic evaluation of flow-cytometric expression of MFI for **j** GPIb (CD42b $n = 4;5;3;3;4$) **k** P-selectin (CD62P, $n = 4$/group) and **l** systematic evaluation of PNCs in % by flow cytometry in the AAR of animals injected with anti-Sema7a or IgG control at 1 and 120 min ($n = 5;6;5;4;4$). Comparisons in **b**, **d** were analyzed by unpaired two-tailed Student's $t$-tests (data are mean ± SD). For **f–l** we used log transformation of data to conform normality. For log-transformed data, unpaired two-tailed Student's $t$-tests were performed on the log values and results are displayed as geometric means and their 95% confidence intervals. $^*p < 0.05$, $^{**}p < 0.01$, and $^{***}p < 0.001$ as indicated.

and are largely unknown. On the other hand, GPIbα binds a large number of immobilized as well as soluble ligands[12–14,24], and it is not clear how they may differentially trigger specific platelet functions. We established soluble Sema7a as an interaction partner of GPIbα that promotes both the thrombotic as well as the thrombo-inflammatory activity of the cells. We could not identify the underlying signaling pathways downstream of GPIb-IX-V, but it appears that the presence of shear flow is required for this pathway to be initiated, indicating that the mechanosensing activity of the receptor may be modulated by Sema7a binding. Inhibition of the ligand-binding domain of GPIbα has been identified as a potential therapeutic strategy to interfere with acute thrombotic disease states as well as the thrombo-inflammatory sequelae in ischemic brain infarction[15]. In a recent study, we further demonstrated that GPIbα inhibition by injection of Fab fragments of the function-blocking antibody p0p/B profoundly reduced immune cell infiltration in a myocardial IR model but did not significantly reduce infarct size. Of note, in that study the treatment was administered after initiation of

reperfusion to avoid excessive surgery-related bleeding[25]. In the current study, we used a different surgical protocol and did not use the same p0p/B antibody approach to interfere with platelet activity during reperfusion. Instead, we used animals lacking a functional GPIb receptor through genetic deletion and found markedly reduced MIRI, and this was not altered by the injection of Sema7a. In vitro, Sema7a strongly enhanced thrombus formation under flow, and this effect was completely reversed by GPIb inhibition. This strongly suggests that Sema7a mediates its action through GPIb, and our immunoprecipitation experiments indicated a direct interaction of the two molecules in blood and myocardial tissue following MIR. Certainly, to some extent Sema7a also exerts its effect through the activation of neutrophils. Yet given the data of the tissue-specific deletion of Sema7a and the experiments in *GPIb-IL-4tg* animals the predominant role of Sema7a during reperfusion injury is mediated by the effect Sema7a on platelets[19]. Further studies will be required to characterize the interaction in more detail and to understand how it exerts its pro-thrombo-inflammatory effects.

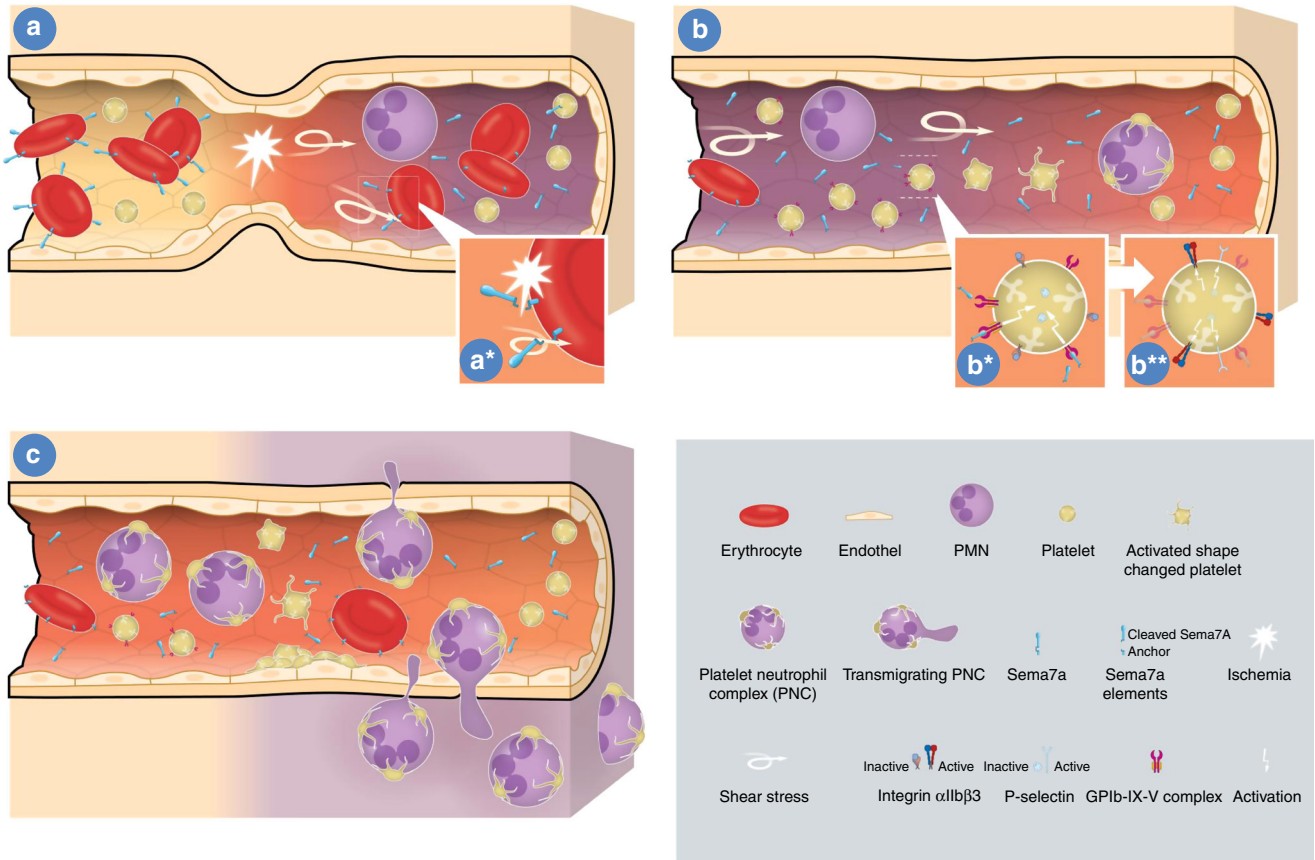

**Fig. 7 Schematic drawing of the role of Sema7a during myocardial IR. a** During myocardial ischemia shear stress and hypoxia result in cleavage of Sema7a from the surface of erythrocytes as the main source of Sema7a within the vascular bed. **b** The released Sema7a then engages the Glycoprotein Ib receptor and activates platelets which then exposes the integrin receptors resulting in platelet–neutrophil complex formation (PNCs). **c** Activated platelets and PNCs migrate from the vasculature to the ischemic tissue which results in tissue injury and destruction.

Sema7a is expressed in several tissues in humans and mice and is rapidly released from the cell membrane during hypoxia. Sema7a stimulates cytoskeletal reorganization in melanocytes and monocytes, which translates into cell morphology changes that can result in the spreading and migration of these cell types[18,26,27]. Sema7a can enhance inflammation in the course of lung injury and tissue hypoxia through action on endothelial cells and T cell-dependent cytokine production[19,20,28]. Here, we have demonstrated that RBC-derived Sema7a is crucial for the described thrombo-inflammatory platelet activity in the setting of experimental MIRI. Sema7a is expressed on human RBCs and is known as the John Milton Hagen blood group antigen; it is released from the cells very rapidly and is present as free Sema7a within plasma[29]. RBCs influence hemostasis and pathological conditions by several known mechanisms: through the supply of ADP, ATP, and nitric oxide[30,31]. Whether Sema7a holds a direct impact on erythrocytes is not described to date. More recent data have shown that RBCs can contribute to the generation of thrombin[32]. However, little is known about a direct influence of RBCs on platelet function. A recent study by Klatt et al. proposed that RBCs contribute to thrombus formation by the FasL/FasR (CD95)-dependent interaction with platelets[33]. There is no evidence that the Sema7a effect reported here is linked to this process or interacts with it. Our data rather show that Sema7a released from RBCs promotes thrombus formation and thrombo-inflammatory myocardial injury through its interaction with GPIb.

Currently, there is no specific treatment available that targets this thrombo-inflammatory myocardial reperfusion injury.

Myocardial cell death during this phase is mediated by a combination of excess inflammation, the presence of thrombi in the coronary microcirculation, metabolic modulation, and electrolyte imbalances. The infiltration of inflammatory cells into the previously ischemic myocardium is a key component of this process, and the interaction of platelets with neutrophils during this phase enhances the inflammatory reaction. We and others have previously shown that the formation of PNCs aggravates inflammatory tissue injury, especially during myocardial reperfusion[8,9]. The infiltration of these inflammatory cells and PNCs into the MI zone leads to elevated concentrations of reactive oxygen species, cytokine release, and the activation of apoptotic and necrotic cell death pathways. This process occurs rapidly, within minutes after the initiation of reperfusion. The expression of adhesion molecules, which act as chemoattractants for neutrophils, and the complement cascade are activated during this phase. In addition, activated platelets release adhesion proteins, growth factors, chemokines, interleukins, and coagulation factors into the local environment. This further alters the chemotactic, adhesive, and proteolytic properties of endothelial cells and promotes the chemotaxis of inflammatory cells to the site of inflammation. We show here that Sema7a strongly increases the formation of PNCs, which are then translocated into the myocardium, where they enhance the above-described effects. Inhibition of Sema7a action through the injection of anti-Sema7a antibody significantly ameliorated this process and resulted in reduced myocardial cell death. We also showed that Sema7a did not directly induce cell death in cardiac myocytes, since this would likely involve caspase-3 activation, which we did not see following exposure of human

myocytes to Sema7a. Our results strongly point to the action of Sema7a on platelets, its interaction with GPIb, formation of PNCs and increased inflammatory tissue injury in the myocardium through this process.

In summary, Sema7a has a significant impact on myocardial ischemia/reperfusion injury. Although Sema7a has no direct action on myocardial cell death, erythrocyte-derived Semaphorin 7A markedly promotes PNC formation and thrombo-inflammation through platelet GPIb. This process enhances myocardial cell death through activated inflammation. Interference with the Sema7a–GPIb axis could be a potent strategy to inhibit this process and significantly reduce myocardial IR injury.

## Methods

**Ethic statement**. Animal protocols were in accordance with the German guidelines for use of living animals and were compliant with all ethical regulations and approved by the Institutional Animal Care and the Regierungspräsidium Tübingen and Würzburg, and the Landesamt für Verbraucherschutz Niedersachsen.

Approval for human sample processing was obtained by the ethics committee of the University of Tübingen (Institutional Review Board). Samples of patients with MI were obtained at presentation to the catheter laboratory and processed (Biobank: 266/2018BO1; Sema7a subanalysis: 266/2018BO2; Clinicaltrial.gov: NCT01417884). Patient samples before and after cardiac surgery were collected as part of the TüSep-Study (NCT02692118). Written informed consent was obtained from each patient before samples were taken. All ethical regulations were complied with.

**Processing of human blood samples**. Human blood samples were taken during coronary intervention, at the end of cardiopulmonary bypass or during occlusion of the coronary arteries during off pump cardiac surgery and processed for flow cytometry. In addition, blood was centrifuged to obtain plasma samples, which were then stored and measured. In another set of experiments, human whole blood withdrawal from healthy donors (Ethics approval 507/2017BO1) was focused on erythrocytes population by flow cytometry and SEMA7A expression was assessed with 1:500 incubation with anti-human SEMA7A monoclonal antibody conjugated with Allophycocyanin (R&D Systems #FAB20681A Minneapolis, USA).

**Mice**. $Sema7a^{-/-}$ mice were generated, validated, and characterized[18]. The corresponding WT controls were bred as littermates of the $Sema7a^{-/-}$ mice. In a subset of experiments, a newly generated Sema7a floxed mouse line ($Sema7a^{loxP/loxP}$/Ozgene) on a C57BL/6 background was crossbred with the following Cre recombinase-positive mouse lines to obtain tissue-specific gene deletion: erythrocyte-specific HbbCre+, myocardial cell-specific Myh6Cre+; endothelial cell-specific Tie2Cre+; and immune cell-specific LysMCre+. In the experiments, tissue-specific gene deleted Sema7a mouse lines ($Sema7a^{loxP/loxP}HbbCre+$; $Sema7a^{loxP/loxP}Myh6Cre+$, $Sema7a^{loxP/loxP}Tie2Cre+$; and $Sema7a^{loxP/loxP}LysMCre+$) were used. $Sema7a^{loxP/loxP}$Cre-negative (−) littermates were used as controls. In a subset of experiments, we used a functional GPIb-knockout mouse line ($GPIb-IL4-tg$) to test the interference of Sema7a with the GPIb receptor[34]. In a subset of experiments $Sema7a^{loxP/loxP}HbbCre+$ animals were reconstituted with recombinant mouse Sema7a (rmSema7a; R&D SYSTEMS, Minneapolis, USA) or recombinant mouse IgG$_{2A}$ Fc (rmIgG$_{2A}$ Fc; control).

**Murine myocardial ischemia and reperfusion model**. This animal model is described in detail elsewhere[35]. Subsets of animals received either recombinant mouse Sema7a (rmSema7a) or recombinant mouse IgG$_{2A}$ Fc (rmIgG$_{2A}$ Fc; control) before the start of experiments or Sema7a antibody (abcam ab23578, Cambridge, UK; anti-Sema7a) or, as control, rabbit IgG sc-2027 (Santa Cruz Biotechnology, Santa Cruz, USA) 5 min before the onset of reperfusion intravenously. For a detailed description see Supplementary material.

**Immunohistochemical techniques**. For immunohistochemical staining, the Vectastain ABC Kit (Linaris, Wertheim, Germany) was used. After inhibiting the nonspecific-binding sites with avidin blocking solution (Vector), the sections were incubated 1:900 anti-CD41 primary antibody (rabbit anti-mouse CD41, abcam ab63983, Cambridge, UK) overnight at 4 °C. Tissue sections were then incubated with biotinylated anti-rabbit IgG for 1 h followed by Vectastain ABC reagent for 30 min, then developed via DAB substrate. Later, PMN were stained with 1:1000 with a rat anti-mouse Ly6B.2 primary antibody (BioRad, MCA771GA, Basel, Switzerland) and HistoGreen as substrate (Linaris, Wertheim, Germany). Counterstaining was performed using nuclear fast red (Linaris, Wertheim, Germany). Histological sections were analyzed for the presence of PNCs by manual count within three independent tissue sections of each animal at magnification ×400.

**Troponin I measurement**. The troponin blood plasma levels of probes taken by central venous puncture after 120 min of reperfusion was measured using the ELISA Kit SEA478Mu (Cloud-Clone Corp., Houston, USA) for murine troponin I type 3 (TNNI3).

**Caspase 3 staining and Caspase 3 ELISA**. Human cardiac myocytes (HCM-c, primary cell line, order-number: C-12810; lot-number: 9083205.4, PromoCell, Heidelberg, Germany) were grown to confluence on chamber slides followed by 6 h of stimulation with rhSEMA7A, rhIgG$_1$ Fc, BSA, or staurosporine (Sigma-Aldrich, Munich, Germany), all 1 µg/ml. After fixation, cells were stained with 1:100 rabbit polyclonal anti-caspase3 antibody (abcam #ab44976 Cambridge, UK). For a detailed description see Supplementary material.

**Sema7a ELISA**. Sema7a ELISAs were performed according to the manufacturer's instructions using the ELISA Kit SEB448Hu for human and SEB448Mu for murine Sema7a (Cloud-Clone Corp., Houston, USA).

**RT-qPCR**. For RNA extraction, we used the peqGOLD TriFast™ (Peqlab; Germany; Erlangen) following the manufacturer's instructions. The iScript kit from Bio-Rad (Bio-Rad; Germany; Munich) was used for cDNA synthesis. Semiquantitative analysis of murine Sema7a was performed by real-time PCR using the sense primer 5′-GTG GGT ATG GGC TGC TTT TT-3′ and the antisense primer 5′-CGT GTA TTC GCT TGG TGA CAT-3′. The reference gene was the murine 18S rRNA gene, with the following set of primers: sense 5′-GTA ACC CGT TGA ACC CCA TT-3′ and antisense primer 5′-CCA TCC AAT CGG TAG TAG CG-3′.

**Protein analysis**. Murine tissue was homogenized and resuspended in RIPA buffer. Probes were separated in SDS–polyacrylamide gels and blotted on PVDF membranes. The following antibodies were used in murine samples: 1:200 anti-Sema7a antibody (abcam #ab23578, Cambridge, UK) and, for the control of loading conditions, 1:300 GAPDH antibody (Santa Cruz Biotechnology #sc-25778, Santa Cruz, USA). For human samples, we used 1:200 polyclonal goat anti-Sema7a antibody (R&D Systems #AF2068, Minneapolis, USA) and 1:1000 β-Actin antibody (Santa Cruz Biotechnology #sc-130656). Bands were detected through chemiluminescence reaction of HRP-conjugated antibodies and developed with Luminol reagent (Santa Cruz Biotechnology #sc-2048).

**Coimmunoprecipation and immunoblotting**. Coimmunoprecipation (Co-IP) was performed according to the manufacturer's instructions using the Pierce Co-IP Kit (cat. no. 261498, Thermo Fisher Scientific, Waltham, USA). Briefly, murine samples were taken after 60-min ischemia followed by 1-min (blood) or 15-min (heart tissue, AAR) reperfusion. Then, 250 µl citrate + blood was lysed in 1 ml IP-Lysis/wash buffer and kept on ice. AAR was incubated in 1 ml IP-Lysis/wash buffer and homogenized in a Precellys 24 (VWR/Peqlab, Erlangen, Germany) and kept at 4 °C for 60 min. All samples were centrifuged at 13,000 × g for 10 min. Total protein of lysates was measured by the Pierce™ BCA Protein Assay Kit (Thermo Scientific, #23225) and analyzed in an Infinite® M200 Pro Plate Reader (Tecan, Männedorf, Switzerland). Ten micrograms of mouse monoclonal antibody against Sema7a (Santa Cruz Biotechnology #sc374432, Santa Cruz, USA) or 10 µg of non-commercialized[36] rat monoclonal antibody against GPIb (p0p4) were immobilized on the Amino Link Plus Coupling Resin. As IgG control, 10 µg rat IgG (sc2016; Santa Cruz Biotechnology) or mouse IgG (X0943; Dako, Glostrup, Denmark) was used. For protein analysis, 30 µl per Co-IP eluate was applied to SDS–PAGE. For immunodetection, a rabbit polyclonal antibody against Sema7a (sc135263; Santa Cruz Biotechnology) and the non-commercialized[37] monoclonal antibody (p0p5) against GPIb were employed. Species-matched alkaline phosphatase-conjugated secondary antibodies were used (Santa Cruz Biotechnology, goat anti rabbit-IgG-AP; #sc-2007, and Thermo Fisher Scientific goat anti rat-IgG-AP; #A18868;). Protein detection was performed using a BCIP/NBT substrate.

**Cardiac magnetic resonance imaging (MRI)**. Animals were subjected to cardiac MRI at 22 weeks of age. Analysis was performed on a clinical workstation with semi-automated contour-tracing software (CVI42, Release 4.1.8 (201), Circle Cardiovascular Imaging Inc., Calgary, Canada). For a detailed description see Supplementary material.

**Flow chamber experiments**. Platelet adhesion under flow was measured by perfusion of murine whole blood on collagen-coated cover slips (200 µg/ml fibrillar type I collagen) at 1000 or 400 s$^{-1}$, as indicated. Platelets were labeled with 0.2 µg/ml of DyLight 488-conjugated anti-GPIX Ig derivative (Emfret Analytics, #M051-1) and treated with rmSema7a or IgG$_{2A}$ Fc for 5 min at 37 °C prior to perfusion. In the case of GPIb-blocking antibody treatment, 100 µg/mouse p0p/B Fab[38] was injected intravenously 20 min before blood was taken for the experiment. A more detailed description of the flow chamber methods is given in Deppermann et al. [39]. In further experiments, blood of functional GPIb ($GPIb-IL4-tg$) animals was used. For further information please see Supplementary material.

**Flow-cytometric analysis.** For flow-cytometric analysis, the following antibodies were used and prepared freshly 1:100 before each experiment: rat anti-mouse Ly6G labeled with BV421 (Biolegend, #127628), rat anti-mouse CD42b-FITC (Emfret, #M040-1), rat anti-mouse CD62P labeled Alexa Fluor 647 (BD Pharmingen, #563674), rat anti-mouse-activated GPIIb/IIIa-PE (Emfret, clone JON/A, #M023-2). For further information please see Supplementary material.

**Gating strategy.** After staining, sample acquisition focused granulocytes by their granularity and surface expression of lymphocyte antigen 6 complex, locus G (Ly-6G), noted as SSC/Ly-6G$^+$. The presence of platelet surface marker CD42b on the surface of SSC/Ly-6G$^+$ events distinguished platelet–neutrophil complexes SSC/Ly-6G$^+$/CD42b$^+$ (PNCs) from free circulating PMNs SSC/Ly-6G$^+$/CD42b$^-$. These two populations were tested for their display of surface transmembrane glycoproteins P-selectin (CD62P) and activated GPIIb/IIIa (clone JON/A). This gating strategy was adopted to analyze peripheral granulocytes in whole blood as depicted in Supplementary Fig. 5a, or to focus cardiac muscle AAR granulocytes as shown in Supplementary Fig. 5b.

**SEMA7A cleavage from erythrocytes.** Erythrocytes were separated from human blood samples using MACS beads (MicroBeads Kit, Milenyi Biotec, Germany). Cells were quantified in a Neubauer chamber. $8 \times 10^8$ erythrocytes were employed per sample. Shear stress was induced by drawing and pushing the erythrocytes through a 27 gauge needle. For hypoxia exposure cells were placed in hypoxic PBS-medium exposed to hypoxia (8% $O_2$) in a Invivo$^2$ 400 hypoxia workstation (Ruskin Technology Ltd; Leeds). At the end of each experiment samples were centrifuged 150×$g$ for 10 min at RT and SEMA7A concentration was measured in the supernatant using the ELISA Kit SEB448Hu.

**Intravital microscopy.** For a detailed description see Supplementary Video 1.

**Data analysis.** All data analysis was done in collaboration with the Institute of Clinical Epidemiology and Applied Biometry of the University Tübingen. Data are usually presented as bar graphs using mean ± SD. Normal distribution was checked using skewness. Raw data are accessible in this publication's data repository. Because the testing for normality may easily lead to statistically none-significant yet meaningless results when sample sizes are small, as was the case throughout this study, we also performed a visual inspection of the data distributions using histograms and tried log-transformation of data to conform to normality, whenever this inspection revealed skewed distributions. For log-transformed data, tests were performed on the log values; data are displayed on a logarithmic scale, showing geometric means and their 95% confidence intervals. Altogether, statistical testing was performed using Student's $t$-tests when comparing two groups; for comparison of multiple groups, we performed one-way analyses of variance followed by Dunnett's tests. For Supplementary Fig. 2: The SEMA7A% on human erythrocytes and age as input variables was analyzed by linear regression analysis and displayed pointwise with 95% confidence bands using healthy humans as donors and age as input variables. For all performed comparisons $p$-value are displayed and $p$-values $< 0.05$ were considered statistically significant, displayed as $p < 0.05$ (∗); $p < 0.01$ (∗∗) and $p < 0.001$ (∗∗∗).

**Reporting summary.** Further information on research design is available in the Nature Research Reporting Summary linked to this article.

## Data availability

The source data underlying main and supplementary figures are provided as a source datafile. The datasets generated during and/or analyzed during the current study are available from the corresponding author on reasonable request.

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

## Acknowledgements

We thank Prof. Martin Eichner, Institute of Clinical Epidemiology and Applied Biometry, Medical Faculty, University of Tübingen, Germany for statistical counseling and work up of the results. We thank Mathias Emde from Emde Grafik (https://www.emde-grafik.de) for help with the preparation of Fig. 7. This work was supported by a grant from the Deutsche Forschungs-gemeinschaft DFG-RO 3671/6-2 (to P.R.) and DFG CRC/TR 240 "Platelets—Molecular, cellular and systemic functions in health and disease" (Project # 374031971) TP B07 (to P.R. and B.N.).

## Author contributions

D.K. performed experiments, analyzed data, wrote parts of the manuscript; T.G. performed experiments, analyzed data, wrote parts of the manuscript; J.V. performed experiments, analyzed data; M.K. performed experiments, analyzed data; H.F.L. collected human samples, analyzed data; G.H. performed murine cardiac imaging, analyzed data; E.L. performed murine cardiac imaging, analyzed data; T.B. collected blood samples and analyzed SEMA7A expression; To.Ge. collected human samples, analyzed data; C.E. performed experiments, analyzed data; H.A.H. collected human samples, analyzed data; B.N. designed research, wrote parts of the manuscript; P.R. designed overall research plan, wrote the manuscript.

## Competing interests

The authors declare no competing interests.
