## [Peer Review File · Nature Communications]

Reviewers' comments:

Reviewer #1 (Remarks to the Author):

Kohler et al focus on a clinically highly relevant topic, cardiac ischemia/reperfusion injury. An area where several clinical trials have failed to establish any successful therapeutic approach. Research on identifying potential novel therapeutic approaches and novel mechanisms is clearly commendable.

The authors started with the observation that the number of platelet-neutrophil-complexes (PNCs) is increased in the blood of patients with acute myocardial infarction (MI) and they also see high levels of SEMA7A in patients with MI. Injection of recombinant SEMA7A causes an increase of infarct size in a temporary LAD ligation mouse model and also a time-dependent increase of PNCs both in blood and ischemic cardiac tissue. The authors further continue with their systematic approach using a respective knock out mouse and seeing the reverse of the effects seen with the recombinant protein injection. Next using cell type specific SEMA7A knock out mice, the authors identified the central role of red blood cells in the LAD temporary ligation model. Mechanistic experiments then performed by the authors showed that SEMA7A did not activate platelets directly but increased binding of platelets to collagen under flow. It was then shown that this effect is dependent on the platelet receptor GPIIb/IIIa. Finally, the authors show that injection of a SEMA7A blocking antibody can indeed reduce infarct size.

Overall this manuscript addresses a clinically very important issue, it reports novel data and includes an impressive scope of experiments. The text is very well written and the data are well introduced, presented and discussed. The authors present a clear-cut rationale and the build-up of the experiments are very logical and well thought through. The experiments performed try both to provide mechanistic explanations and also give a therapeutic translational perspective.

Major mechanistic questions:

The authors present strong data indicating that red blood cells are the source of SEMA7A. How do the authors envisage the role of red blood cells? Do they 'become' ischemic or do they adhere at the ischemic endothelium? What induces the release of SEMA7A? This is an important part of the overall story and to this reviewer it seems that this part has not been investigated at all. Can the authors provide data (red blood cell ischemia, activation, adhesion or similar) and a mechanistic concept for the role of red blood cells in cardiac ischemia/reperfusion?

The role of neutrophils is similarly unclear, although they play a central role in the authors' introduction. Is there a direct effect of SEMA7A on neutrophils? Are neutrophils necessary for the effects seen by the authors? Are they just a measure of platelet activation? Are platelet-neutrophil complex numbers just a read out for platelet activation? There are reports that platelets on their own are accumulating in ischemic areas and that their therapeutic targeting can reduce cardiac ischemia/reperfusion injuries. These points need a clearer discussion of the literature and the overall mechanistic concept needs to include the role of platelets and neutrophils as pathological drivers of ischemia/reperfusion injury.

The authors should consider adding a schematic drawing, describing the source of SEMA7A, the mechanisms of release and its effects on specific cell types and the overall effects.

Figure 1: The exact statistical tests used need to be listed in the legends of the figures.

The authors would preferable have a control group that is gender and particularly age matched. This is a major limitation that should be ideally addressed by a better control group or need to be discussed at least as a major caveat.

Is the plasma level of SEMA7A age dependent?

At what time points were the levels in humans determined?

The authors state that the number of PNCs correlated with increased levels of the soluble neuronal guidance protein semaphorin 7A (SEMA7A). The authors might use a different wording here as they did not really investigate a correlation between the two variables but rather found increases of both in patients with MI.

Figure 2: Can the authors provide more details on the recombinant rmSem7a? This reviewer could not find the protein in a search of products of R&D. Also more details on the control (emIgG2a-FC should be listed to clearly understand that this is the correct control.

Figure 5: Some of the figure labels are distorted.

Figure 6: How much of the blocking antibody and also its control was injected? The authors need to list a the dose, preferentially in the result section, legend or both.

Suppl Figure 4.: How did the authors define the gate threshold for CD42b? This gate looks quite artificial especially in B. Can the authors show a platelet population in the flow cytometric setting used? In the main figures the authors seem to use a different threshold for CD42b as well. Please clarify.

Reviewer #2 (Remarks to the Author):

In the presented manuscript Köhler and colleagues report about their investigations on SEMA7A in the context of cardiac ischemia reperfusion injury. They start off with analysis in human samples where they find elevated levels of platelet-neutrophil complexes (PNCs) in patients with acute MI, compared to controls or from patients undergoing cardiac surgery. In addition, they also find SEMA7A elevated in this group of patients. Administration of recombinant SEMA7A led to increased infarct size compared control treated animals and higher numbers of PNCs accumulating at the lesion site. In contrast, SEMA7A knockout mice presented with smaller infarcts compared to their littermates. In order to evaluate the cell-specific contribution of SEMA7A after MI, several transgenic *Sema7a loxP/lox -cre* mice were generated and it was found that in particular red blood cells and (to a lesser extend) endothelial cells contribute to SEMA7A Production. Further, the authors present elegant in vitro and in vivo experiments that demonstrate an interaction of SEMA7A with glycoprotein Ib on platelets. To test if blockade of SEMA7A exhibits beneficial after myocardial IR, the authors injected a function-blocking anti-SEMA7A antibody and compared it to IgG control treated animals. Anti-SEMA7A administration blunted the generation of PNCs and resulted in reduced infarct size compared to control-treated animals.

Overall, this is a very interesting, well designed study. The manuscript is well written and the concept is novel.

Could the authors speculate on what specifically triggers red blood cells to release SEMA7A only in response to myocardial infarction?

Are there also functional (e.g. in LV-EF) differences upon SEMA7A-administration or -blockade?

When plotting SEMA7A vs. PNCs, is there a correlation between higher levels of SEMA7A and increased numbers of PNCs in these patients?

Does administration of Glycoprotein IIb/IIIa inhibitors (e.g. Tirofiban) lead to reduced numbers of PNCs?

The labelling of graphs in figures 2d, 6b and 6c are not readable.

Reviewer #3 (Remarks to the Author):

This manuscript examines platelet-neutrophil interaction responsible for reperfusion injury after myocardial ischemia. Recently, there are extensive literature implicating roles for neutrophil extracellular traps and platelet-neutrophil interaction in inflammatory tissue injury, but the its underlying mechanisms that regulate platelet-neutrophil interaction are still being established. The authors demonstrate that Sema7a increased in the plasma of MI patients but not in any of the other tested patient groups. Sema7a null mice have reduced myocardial damage following in vivo ischemia. Sema7a derived from red blood cell is central to myocardial damage. Sema7a interacts with platelet glycoprotein Ib producing vWF receptor complex GPIb-IX-V in flow-dependent manner. The pathophysiological significance of this pathway is demonstrated by the fact that Sema7a loxP/loxP Hbb Cre⁺ mice, but not Sema7a loxP/loxP Myh6Cre⁺, and have reduced myocardial damage and anti-Sema7a treatment reduces platelet-neutrophil complexes (PNCs) formation and reperfusion injury after myocardial ischemia. This paper contains a lot of data which support the major conclusions of the paper. Given the interest in platelet-neutrophil interaction in inflammatory tissue injury and the novelty of the data. This study is interesting. However, there are multiple concerns that will need to be addressed.

1) Sema7A is expressed on activated T cells. Sema7A is well known to bind to alpha1beta1 integrin, alpha v beta 1 integrin, or plexin-C1 as receptor in monocytes and macrophages. Do these receptors express in monocytes and macrophages in myocardial IR injury? Authors suggested infarct size was unaltered in Sema7a loxP/loxP LysMCre⁺ mice compared to controls. How do you think about the significance of $\alpha 1\beta 1$ integrin, $\alpha v\beta 1$ integrin, or plexin-C1 in myocardial IR injury?

2) Authors need to determine if the involvement of these receptor in myocardial IR injury, such as studies using $\alpha 1$ integrin-deficient or plexin-C1-deficient mice, or anti-integrin-Ab, anti-plexin-C1 Ab.

3) Perhaps the most novel aspect of the study is the identification of red blood cell as origin of Sema7A expression. Sema7a loxP/loxP Hbb Cre⁺ mice have reduced myocardial damage. But they need to determine if the reduced myocardial damage of Sema7a loxP/loxP Hbb Cre⁺ mice is reversed by administration of recombinant Sema7A protein or blood transfusion of red blood cells derived from wild type mice. Additional experiments could solidify the conclusion

4) Semaphorin 7A was previously reported to contribute tissue fibrosis including lung and liver. Does Sema7a participate in fibrosis after myocardial infarction?

5) The authors show significantly smaller infarcts in *Sema7aloxP/loxP Tie2Cre+* mice, compared to controls, accompanied by a reduced number of PNCs. How do you think about the significance of the reduced myocardial damage in *Sema7aloxP/loxP Tie2Cre+* mice? Does this result in *Sema7aloxP/loxP Tie2Cre+* mice reflect an involvement of the vWF receptor complex GPIb-IX-V in this process?

6) It is very hard to see dot plots in flow cytometric studies with figures this small. should be enlarged

7) Result portion: line 14, extracorporeal?

Revision of NCOMMS-19-10591-T

„Red Blood Cell (RBC)–derived Semaphorin 7A promotes thrombo-inflammation in myocardial ischemia-reperfusion injury through a platelet GPIb-dependent mechanism”

and point to point answers.

Reviewer #1:

The Reviewer states that the authors present strong data indicating that red blood cells are the source of SEMA7A. How do the authors envisage the role of red blood cells? Do they ‘become’ ischemic or do they adhere at the ischemic endothelium? What induces the release of SEMA7A? Can the authors provide data (red blood cell ischemia, activation, adhesion or similar) and a mechanistic concept for the role of red blood cells in cardiac ischemia/reperfusion? We thank the reviewer for these points raised, and have provided experimental data about the release of SEMA7A from erythrocytes in the revised manuscript (see Supplemental Figure 4). These demonstrate that hypoxia and sheer stress are the mechanism by which SEMA7A is released from erythrocytes, both of which are present during periods of ischemia. To give the reader a better understanding of this we have also provided a scheme that describes the role of Semaphorin 7A during ischemia reperfusion injury. This is provided in Figure 7.

The Reviewer states that the role of neutrophils is similarly unclear, although they play a central role in the authors' introduction. Is there a direct effect of SEMA7A on neutrophils? Are neutrophils necessary for the effects seen by the authors? Are they just a measure of platelet activation? Are platelet-neutrophil complex numbers just a read out for platelet activation? There are reports that platelets on their own are accumulating in ischemic areas and that their therapeutic targeting can reduce cardiac ischemia/reperfusion injuries. These points need a clearer discussion of the literature and the overall mechanistic concept needs to include the role of platelets and neutrophils as pathological drivers of ischemia/reperfusion injury. We thank the reviewer for this point raised, we have therefore performed additional experiments to provide insight into the effect of Semaphorin 7A on neutrophils. We have demonstrated these in Supplemental Figure 7. This shows that Semaphorin 7A increases the adhesion and migration of neutrophils into

the ischemic tissue, and such increases neutrophil activation, therefore the effect of Semaphorin 7A is not only mediated by the activation of platelets, however given the results in the conditional KO animals this is the predominant effect. We have shown the effect of Sema7a during hypoxia in the past also in our work (Morote-Garcia 2012, PNAS). This work shows that Sema7a can induce neutrophil migration during periods of tissue hypoxia, and certainly the activation of neutrophils enhances the PNC formation.

Yet the effect described here is predominantly mediated by action of Semaphorin 7A on platelets, when looking at Figure 5a, c, e and f. In addition, to completely work up the role of Semaphorin 7A on neutrophils this would also need deeper work into the role of Semaphorin 7A on integrins and plexins, which we are working on at the moment but is also out of the scope of the presented manuscript. But this work will take an additional several months to have a full insight into the role of Semaphorin 7A on all aspects of neutrophil biology, but we are sure that the provide experiments highlight the role of Semaphorin 7A on neutrophils sufficiently.

The Reviewer states that the authors should consider adding a schematic drawing, describing the source of SEMA7A, the mechanisms of release and its effects on specific cell types and the overall effects. We thank the reviewer for this point and have included a scheme about the role of Semaphorin 7A during myocardial IR injury and included this into Figure 7.

The Reviewer states that in Figure 1: The exact statistical tests used need to be listed in the legends of the figures. We thank the reviewer for this point and have included this information into the revised version of the manuscript.

The Reviewer states that the authors would preferable have a control group that is gender and particularly age matched. This is a major limitation that should be ideally addressed by a better control group or need to be discussed at least as a major caveat. We thank the reviewer for this point, indeed an ideal control group would have sham cardiac catheterization and the blood drawn during this, without myocardial ischemia and without drugs used during MI treatment. The group given in Figure 1 was shown to described the

baseline value of SEMA7A in healthy individuals. We believe that the labeling is distracting and have changed this into healthy. But to further work this up we have provided additional data from different age groups in work with the Department of Transfusion Medicine and are able to show that SEMA7A is evenly distributed in its expression on the surface of erythrocytes in all age groups and have included this information into the revised version of the manuscript in Supplemental Figure 2.

The Reviewer states that: Is the plasma level of SEMA7A age dependent? We have investigated this and can demonstrate that this is not the case, please see Supplemental Figure 2 for this.

The Reviewer states at what time points were the levels in humans determined? The levels were determined during coronary intervention, at the end of cardiopulmonary bypass or during occlusion of the coronary arteries during off pump cardiac surgery. However, the time point after the onset of chest pain might of course vary but this is due to the fact that patients did not present right away after the onset of symptoms to the emergency department. For clarification we have included a statement into the methods section for a better understanding.

The Reviewer states the authors state that the number of PNCs correlated with increased levels of the soluble neuronal guidance protein semaphorin 7A (SEMA7A). The authors might use a different wording here as they did not really investigate a correlation between the two variables but rather found increases of both in patients with MI. We thank the reviewer for this point, and indeed the point of this was just to show an association of these two. It is correct that the exact correlation was not investigated in our study and the wording was changed accordingly of course, for details please see last paragraph of the introduction.

The Reviewer states figure 2: Can the authors provide more details on the recombinant rmSema7a? This reviewer could not find the protein in a search of products of R&D. Also more details on the control (rmIgG2a-FC should be listed to clearly understand

that this is the correct control. We thank the reviewer for this point, we have clarified this in the manuscript. Please also see

rmSema7a:

https://www.rndsystems.com/products/recombinant-mouse-semaphorin-7a-fc-chimera-protein-cf_1835-s3

rmIgG2a Fc:

https://www.rndsystems.com/products/recombinant-mouse-igg2a-fc-protein-cf_4460-mg

The Reviewer states that in Figure 5: Some of the figure labels are distorted. We thank the reviewer for this point, this must be due to the up-loading process of the file. We hope that this is not the case in the revised version of the manuscript.

The Reviewer states that in Figure 6: How much of the blocking antibody and also its control was injected? The authors need to list the dose, preferentially in the result section, legend or both. We thank the reviewer for this point, we have included this into the manuscript of course in the results section.

The Reviewer states that in Supplemental Figure 4.: How did the authors define the gate threshold for CD42b? This gate looks quite artificial especially in B. Can the authors show a platelet population in the flow cytometric setting used? In the main figures the authors seem to use a different threshold for CD42b as well. Please clarify. We thank the reviewer for this point and have clarified the gating strategy in the method section and also in Supplemental Figure 5 legend. Indeed there was an inconsistency in the gating strategy which has been corrected now

Reviewer #2:

The Reviewer states: could the authors speculate on what specifically triggers red blood cells to release SEMA7A only in response to myocardial infarction? We thank the

reviewer for this comment and have performed additional experiments to address this point. We have included this into Supplemental Figure 4, here we are able to show that shear stress and hypoxia are triggers to release SEMA7A from erythrocytes.

The Reviewer states whether there are also functional (e.g. in LV-EF) differences upon SEMA7A-administration or –blockade? We have not addressed this point specifically, but injection of rmSema7a did not change LV function during experimental procedures and we did not observe such a dynamic change in our experiments, and since the heart is exposed during this procedure the investigator can determine this quite well.

The Reviewer states: When plotting SEMA7A vs. PNCs, is there a correlation between higher levels of SEMA7A and increased numbers of PNCs in these patients? We thank the reviewer for this point yet as stated in the answer to Reviewer 1, as also mentioned by Reviewer 1 we did not thoroughly investigate the correlation between PNCs and SEMA7A in humans, otherwise we would have to take more co-founders into account such as for example the medication before symptom onset. Medication used such as aspirin, clopidogrel and GPIIb/IIIa inhibitors have significant impact on PNC formation. In addition time of onset from symptoms of myocardial infarct to intervention would be important for this, and a better control group such as NSTEMI or patients undergoing acute PCI without myocardial ischemia. We can therefore say that we just looked at an association of Sema7A levels with the presence and activation status of PNCs in humans and have used the best available comparison groups.

The Reviewer states: Does administration of Glycoprotein IIb/IIIa inhibitors (e.g. Tirofiban) lead to reduced numbers of PNCs? The administration of GPIIb/IIIa inhibitors reduced the number of PNCs in this context and was shown by several investigators before, for this please see Köhler et al Circulation 2011, Ikuta Int J Mol Med. 2005 , Straub A Eur J Cardiothorac Surg. 2005 , Salanova B J Biol Chem. 2007 , Li X Tex Heart Inst J. 2009 and Horn M Thromb Haemost. 2012. In the light of this, the patients with myocardial ischemia would have even more PNCs if the medications used before and after the PCI intervention would not reduce the number of PNCs.

The Reviewer states that the labeling of graphs in figures 2d, 6b and 6c are not readable.
We thank the reviewer for this point and have changed this accordingly.

Reviewer #3:

1) Sema7A is expressed on activated T cells. Sema7A is well known to bind to alpha1beta1 integrin, alpha v beta 1 integrin, or plexin-C1 as receptor in monocytes and macrophages. Do these receptors express in monocytes and macrophages in myocardial IR injury? Authors suggested infarct size was unaltered in Sema7a^{loxP/loxP} LysM^{Cre+} mice compared to controls. How do you think about the significance of $\alpha 1\beta 1$ integrin, $\alpha v\beta 1$ integrin, or plexin-C1 in myocardial IR injury? We thank the reviewer for this point, but given the Editors comments we did address this point experimentally. It is right that Sema7a is expressed on different cell types and has been shown to act on several cell types, including monocytes and macrophages. The mentioned receptors are expressed in MIRI, yet a detailed experimental work up would be necessary to address each receptors relevance for MIRI. We believe this is beyond the scope of a revision of a manuscript.

2) Authors need to determine if the involvement of these receptor in myocardial IR injury, such as studies using $\alpha 1$ integrin-deficient or plexin-C1-deficient mice, or anti-integrin-Ab, anti-plexin-C1 Ab. Please see the answer to the point addressed above and the answer to the editor.

3) Perhaps the most novel aspect of the study is the identification of red blood cell as origin of Sema7A expression. Sema7a^{loxP/loxP} Hbb^{Cre+} mice have reduced myocardial damage. But they need to determine if the reduced myocardial damage of Sema7a^{loxP/loxP} Hbb^{Cre+} mice is reversed by administration of recombinant Sema7A protein or blood transfusion of red blood cells derived from wild type mice. Additional

experiments could solidify the conclusion. We thank the reviewer for this point and to clarify this we have performed additional experiments. For his we have reconstituted the Sema7a loxP/loxP Hbb Cre+ animals with rmSema7a and can show that this results in an increase of MIRI, results for this are shown in Supplemental Figure 12.

4) Semaphorin 7A was previously reported to contribute tissue fibrosis including lung and liver. Does Sema7a participate in fibrosis after myocardial infarction? We thank the reviewer for this point. To clarify this, we would have to address the role of Sema7a during the resolution of inflammation and during fibrosis if the heart. The model used in this manuscript does not allow such conclusions, however given the existing literature it is likely that Semaphorin 7A has a role in myocardial fibrosis.

5) The authors show significantly smaller infarcts in Sema7a^{loxP/loxP} Tie2Cre⁺ mice, compared to controls, accompanied by a reduced number of PNCs. How do you think about the significance of the reduced myocardial damage in Sema7a^{loxP/loxP} Tie2Cre⁺ mice? Does this result in Sema7a^{loxP/loxP} Tie2Cre⁺ mice reflect an involvement of the vWF receptor complex GPIb-IX-V in this process? We thank the reviewer for this point. We have shown in the past that SEMA7A expressed on endothelial cells can increase neutrophil migration (Morote-Garcia 2012, PNAS). As such a result as described in the Sema7a^{loxP/loxP} Tie2Cre⁺ mice is not unexpected, since the impact of neutrophils on myocardial IR injury is well described and known. Yet the described role of Sema7a is mediated by the action of Sema7a on the GPIb receptor complex.

6) It is very hard to see dot plots in flow cytometric studies with figures this small. should be enlarged. We thank the reviewer for this point but would leave this up to the journal editorial team to decide whether the presented dot plots are sufficient in size or not.

7) Result portion: line 14, extracorporeal? We thank the reviewer for this point, and have corrected it.

REVIEWERS' COMMENTS:

Reviewer #1 (Remarks to the Author):

The authors have adequately addressed my previous comments.

Reviewer #2 (Remarks to the Author):

Rosenberger and colleagues present a revised manuscript on their work investigating the role of Semaphorin 7A in myocardial ischemia-reperfusion injury. In general, the authors have done a good job in addressing the reviewers concerns.

As pointed out previously by reviewer 3 (Q3) however, an important aspect is “the identification of red blood cells as origin of Sema7A expression” and if the infarct-reducing effect of Sema7A deficiency can be reversed in a rescue experiment. Data addressing this aspect are shown in Supplemental Figure 12. While reversal with recombinant Sema7A administration is compelling, analysis after administration of Sema7A competent red blood cells would have been an even stronger proof. As a further specific comment to Supplemental Figure 12: this reviewer was not able to identify any clearly stained neutrophils in the presented sections (in both examples of SupFig 12c). Size bars are generally lacking in all histology slides. What is also interesting is that Sema7A administration in this experiment resulted in Troponin I levels >2000 pg/ml which is more than double the level of what was detected in the other experiments. Could the authors discuss why that is? In addition, it would be informative why 7 mice per group were used for Troponin measurements, but only 6 for evaluation of infarct size (wrongly labeled in the raw data set file under SupFig14b?)?